# The use of prophylactic intra-aortic balloon pump in high-risk patients undergoing coronary artery bypass grafting

Ken Nakamura[1☯]*, Azumi Hamasaki[2☯], Tetsuro Uchida[2], Kimihiro Kobayashi[1], Ri Sho[3], Cholsu Kim[1‡], Hideaki Uchino[1‡], Takao Shimanuki[1‡], Mitsuaki Sadahiro[2]

**1** Division of Cardiovascular Surgery, Nihonkai General Hospital, Sakata, Japan, **2** Second Department of Surgery, Yamagata University Faculty of Medicine, Yamagata, Japan, **3** Department of Public Health, Yamagata University Faculty of Medicine, Yamagata, Japan

☯ These authors contributed equally to this work.
‡ These authors also contributed equally to this work.
* ken.nakamura622@gmail.com

**Data Availability Statement:** All relevant data are within the manuscript and its Supporting Information files.

**Funding:** We had no funding support.

## Abstract

### Objective

Intra-aortic balloon pump (IABP) is one of the most commonly used mechanical circulatory assist devices for high-risk patients undergoing cardiac surgery. In an effort to validate previously reported clinical outcomes, we describe the preoperative characteristics and outcomes of patients who underwent prophylactic IABP in high-risk patients undergoing coronary artery bypass grafting (CABG).

### Design

A prospective observational study

### Methods

From 2005 to 2017, 471 patients underwent either isolated or combined CABG at our institution. Of those, 393 patients underwent isolated CABG and were included for the analysis. Eighty-five patients (22%) were considered high-risk and underwent prophylactic IABP, with subsequent review of surgical morbidity and mortality rates.

### Results

The 30-day postoperative mortality (prophylactic IABP group vs non prophylactic IABP group: 0% vs 1.6%, $p = 0.589$) and major adverse cardiac or cerebrovascular events (5.9% vs 3.3%, $p = 0.333$) were not significantly different between the two groups. Prolonged mechanical ventilation (>72 hours) (12.5% vs 4.2%, $p = 0.014$) occurred more frequently in the prophylactic IABP group.

**Competing interests:** The authors have declared that no competing interests exist.

## Conclusions

No IABP-related complications were noted, emphasizing that the use of prophylactic IABP in high-risk patients undergoing CABG is an acceptable option.

## Introduction

Patients with ischemic heart disease, particularly those requiring coronary artery bypass grafting (CABG), pose a major challenge for anesthesiologists and cardiothoracic surgeons, emphasizing the need for risk stratification and prophylactic measures to reduce postoperative morbidity and mortality. Cardiopulmonary bypass (CPB) and aortic cross-clamping or off-pump coronary artery bypass (OPCAB) grafting are the most commonly utilized approaches during CABG, but are highly dependent on the patient's condition, the proficiency level of the surgeon and staff, as well as institutional policy.

Nihonkai General Hospital is a community hospital in the coastal area of Yamagata Prefecture, covering a population of 300,000 people across multiple mountainous terrains. Due to the lack of physician-follow up, shortage of doctors, and the challenging geography many high-risk patients, including emergent cases, often present to our institution. However, as a general community hospital, subspecialized providers–such as cardiac anesthesiologists–are unavailable, making preoperative and prophylactic interventions critical amongst patients undergoing intensive procedures.

At present, intra-aortic balloon pump (IABP) is the most commonly used device for circulatory assistance in cardiac surgery and preoperative prophylactic IABP has been shown to improve outcomes in high-risk patients [1, 2]. In an effort to validate prior studies on the benefits of postoperative clinical outcomes, we conducted a prospective observational study of preoperative prophylactic IABP in high-risk patients undergoing CABG with the aim of describing subsequent morbidity and mortality rates.

## Patients and methods

Nihonkai General Hospital institutional ethical review board approved the research protocol prior to initiation of this study, and written consent was obtained from all subjects.

This was a single-center, prospective observational study conducted at Nihonkai General Hospital, and involved 471 unique patients from December 2005 to December 2017 who underwent isolated or combined CABG. Of those, 393 patients underwent isolated CABG and were included for the analysis. Prior to all CABGs, a multidisciplinary team of cardiothoracic surgeons, cardiologists, nurses, and medical technicians met to discuss the indication for preoperative and prophylactic IABP in high-risk patients. High-risk patients were those with hemodynamically stable and meeting two or more of the following criteria: 1) New York Heart Association (NYHA) Functional Class III or IV; 2) left ventricular ejection fraction (LVEF) less than 40% (evaluated by preoperative echocardiography with modified Simpson methods); 3) left-ventricular end-diastolic internal diameter >65mm; 4) left main stenosis >50%; 5) diffuse coronary artery disease, defined as the requirement for three or more distal anastomoses; and 6) refractory unstable angina. Patients with contraindications to IABP, defined as severe peripheral vascular disease, aortic regurgitation, dissection, or aneurysm, were excluded from this study.

Of the 393 unique patients who presented for CABG, 85 (22%) were considered high-risk by the aforementioned definitions and underwent prophylactic IABP (**Fig 1**). Baseline

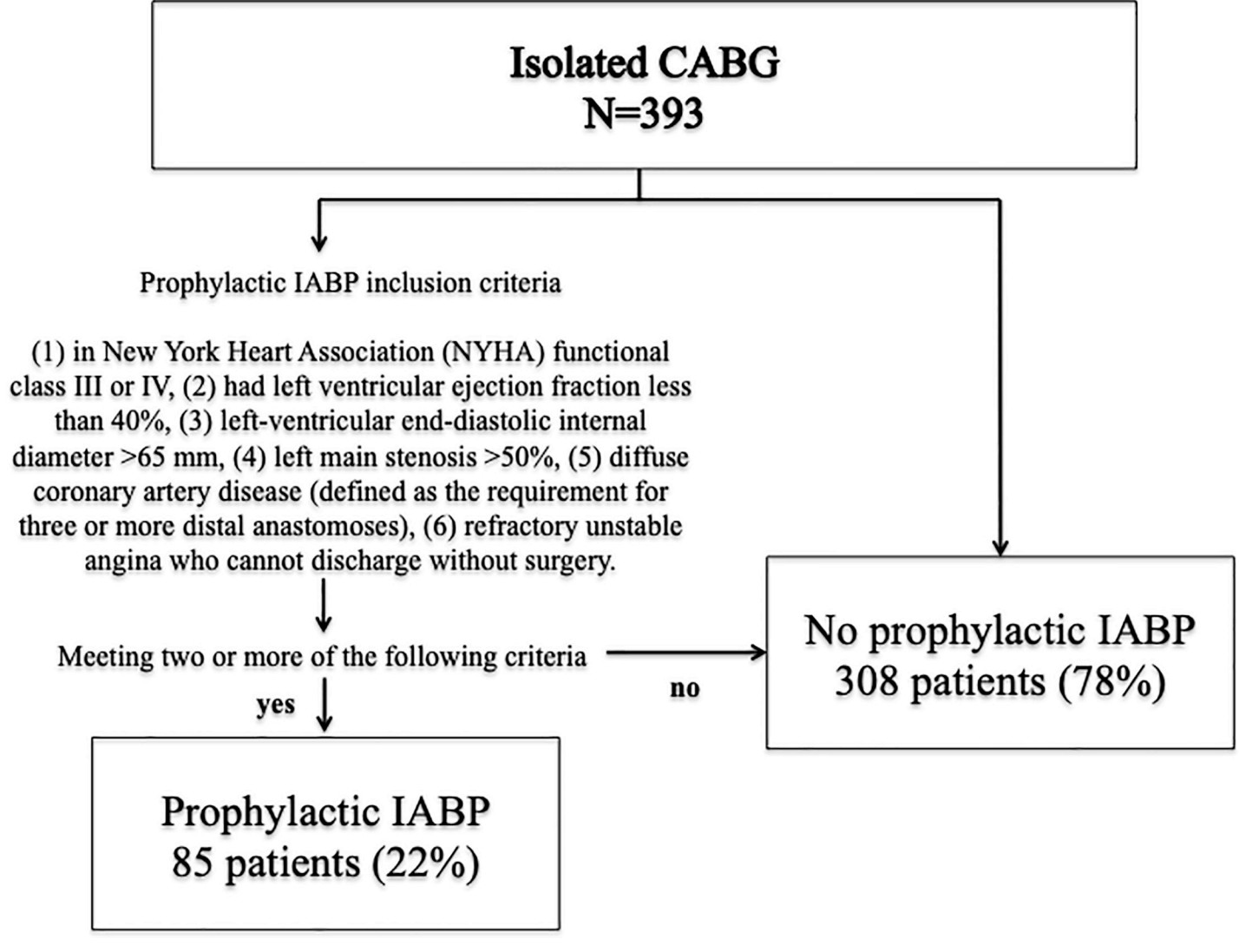

**Fig 1. Summary flow diagram of patient disposition.**

characteristics and intraoperative data were collected and are detailed in **Table 1.** We conducted a cardiac operative risk assessment using European system for cardiac operative risk evaluation (EuroSCORE II), a validated risk prediction model. Acute kidney injury (AKI) was defined by increased serum creatinine (SCr) to 0.3 mg/dl/48h or more, increased SCr by 1.5 times or more, or decreased urine output to 0.5ml/kg or less during 6 hours [3, 4].

All high-risk patients had prophylactic IABP. IABP was inserted in the catheterization lab the day prior to CABG, with continuous intraprocedural IABP, and postoperatively as clinically indicated; patients were monitored in the ICU pre- and post-operatively. Non-high-risk patients received IABP if clinically indicated based on hemodynamic instability pre- or intraoperatively. All patients who received a prophylactic IABP entered the ICU and were waiting.

Placement of IABP was through percutaneous puncture of the femoral artery, with subsequent introduction of an 8-French balloon catheter with a guide wire through an arterial sheath. The IABP balloon was selected according to the height of the patients and then connected to a CS300™ (Getinge AB, Gothenburg, Sweden). Positioning was confirmed

**Table 1. Baseline patient characteristics, preoperative data.**

| Characteristic | Total (n = 393) | Prophylactic IABP (n = 85) | No prophylactic IABP (n = 308) | p-value (prophylactic IABP vs. non-prophylactic) |
|---|---|---|---|---|
| **Age, y** | | | | *0.252* |
| Mean ± SD | 69 ±10 | 70 ± 10 | 68 ± 9 | |
| Median (IQR) | 70 [63, 75.5] | 69 [63, 78] | 70 [63, 75] | |
| **Male, %** | 86.1 | 87.1 | 85.7 | *0.861* |
| **Height, cm** | | | | *0.621* |
| Mean ± SD | 161.2 ± 8.6 | 162.1 ± 8.6 | 161.6 ± 8.4 | |
| Median (IQR) | 162.00 [156.20, 167.00] | 163.00 [157.38, 167.13] | 161.60 [156.00, 167.00] | |
| **Weight, kg** | | | | *0.808* |
| Mean ± SD | 61.2 ± 12.7 | 60.9 ± 12.4 | 61.3 ± 12.7 | |
| Median (IQR) | 60.15 [53.00, 69.43] | 60.50 [54.00, 69.75] | 60.15 [52.85, 69.43] | |
| **BMI, kg/ m2,** | | | | *0.595* |
| Mean ± SD | 23 ± 4 | 23 ± 4 | 23 ± 4 | |
| Median (IQR) | 23.44 [20.63, 25.50] | 23.75 [20.74, 25.62] | 23.39 [20.62, 25.49] | |
| **BMI $\geq$30, %** | 4.7 | 2.4 | 5.4 | *0.383* |
| **OMI, %** | 50.4 | 43.5 | 42.3 | *0.178* |
| **Hypertension,%** | 80.7 | 76.2 | 82 | *0.273* |
| **Hyperlipidemia,%** | 64.9 | 59.5 | 66.4 | *0.247* |
| **Diabetes mellitus,%** | 47.0 | 45.2 | 47.5 | *0.805* |
| **Insulin, %** | 12.2 | 4.8 | 14.2 | *0.022* |
| **Family history,%** | 16.5 | 12.9 | 17.9 | *0.426* |
| **Smoking,%** | 66.5 | 69.4 | 65.6 | *0.644* |
| **Current smoker,%** | 13.2 | 24.6 | 9.4 | *0.003* |
| **Post PCI, %** | 22.4 | 19.1 | 23.4 | *0.462* |
| **PAD, %** | 7.1 | 2.4 | 8.4 | *0.081* |
| **SCr, mg/dl, Mean ± SD** | 1.0 ± 1.0 | 0.8 ± 0.3 | 1.1 ± 1.1 | *0.172* |
| **CRF, %** | 11 | 6 | 12.3 | *0.116* |
| **hemodialysis, %** | 4.3 | 1.2 | 5.2 | *0.137* |
| **Stroke, %** | 8.4 | 7.1 | 8.8 | *0.825* |
| **Recent AMI, %** | 10.9 | 14.1 | 10.1 | *0.326* |
| **NYHA III or IV, %** | 26.5 | 35.3 | 24.1 | *0.051* |
| **LVEF, %** | | | | *<0.0001* |
| Mean ± SD | 55 ± 11 | 50 ± 18 | 54 ± 14 | |
| Median (IQR) | 57.50 [43.00, 67.00] | 50.00 [34.00, 66.0] | 59.00 [46.00, 68.00] | |
| **LMT stenosis, %** | 40 | 53 | 37 | *0.009* |
| **Coronary stenosis $\geq$50, Mean ± SD** | 2.5 ± 0.7 | 2.6 ± 0.7 | 2.5 ± 0.7 | *0.101* |
| **emergency operation, %** | 10.9 | 22.4 | 7.8 | *0.001* |
| **EuroSCORE II, Mean ± SD** | 2.0 ± 2.3 | 2.8 ± 2.9 | 1.7 ± 2.1 | *0.0001* |

SD: Standard Deviation, IQR: Interquartile Range, BMI: Body Mass Index, OMI: Old Myocardial Infarction, PCI: Percutaneous Coronary Intervention, PAD: Peripheral Arterial Disease, SCr: Serum Creatinine, CRF: Chronic Renal Failure, AMI: Acute Myocardial Infarction, NYHA: New York Heart Association, LVEF: Left Ventricular Ejection Fraction, LMT: Left main coronary trunk, EuroSCORE: European system for cardiac operative risk evaluation

immediately with chest angiography. Use of an IABP was discontinued when hemodynamic stability was restored.

IABP weaning was dependent on stabilization of circulatory dynamics. Weaning was done when diuresis was present and there was warm in peripheral sensation. However, there was no numerical target.

All non-urgent patients underwent extensive cardiac rehabilitation prior to surgery. Even patients that were admitted into the emergency room underwent cardiac rehabilitation if surgical intervention was not urgent. Additionally, we treated dental conditions, glycemic imbalances, and co-morbid treatable diseases such as carotid artery stenosis prior to cardiac surgery. Extensive rehabilitation had been especially applied for patients whose activities of daily living (ADL) were limited due to treatment for heart failure after hospitalization. Rehabilitation aims for patients to be as close to the original ADL as possible. As a first goal, we place importance on standing (out-of-bed exercise) [5]. Extensive rehabilitation is not applied to patients with a high risk for preoperative load, such as cases with LMT lesions, symptom cases, and unstable circulation. Additionally, patients who developed symptoms during extensive rehabilitation underwent surgery immediately. The procedures and the distribution were the same in both groups. It had been uniform throughout the study time. There were 10 surgeons during the observation period, 5 of which were staff surgeons and 5 resident surgeons. There were 7 surgeons in prophylactic IABP group(4 staff surgeons and 3 resident surgeons) and 10 surgeons in no prophylactic IABP group. According to the policy of the first staff surgeon, the procedures are consistently the same and the first surgeon (the one that performed the surgery) is currently involved in the treatment as staff. The guidelines for the application of prophylactic IABP, the choice of Off pump / On Pump, and the type of graft used are consistent for all patients. A CPB circuit was used when deemed necessary during the preoperative assessment. It primarily included criteria such as a large left ventricle and low cardiac function. Additionally, on-pump CABG was scheduled following a comprehensive risk assessment, which included the location and quality of target vessels, and accounted for technically challenging cases. If complete revascularization was feasible, OPCAB was scheduled. Conversion to CPB was considered if there was any evidence of hemodynamic instability, such as ventricular arrhythmia, hypotension (systolic pressure $\leq$ 80 mmHg), and cardiac arrest during OPCAB procedures. The OPCAB is performed after a median sternotomy. The heart was displaced using a posterior pericardial stitch, gauze, and a tissue stabilizer (Octopus Evolution tissue stabilizer and Octopus Evolution AS tissue stabilizer; Medtronic Corporation, Minneapolis, MN, USA). Body position changes and gravity support (Trendelenburg, right and left table rotations) were carried during surgery. A $CO_2$ blower mister device was used in situations where a bloodless field was not achieved after proximal target vessel occlusion. An intracoronary shunt (Phycon coronary minishunt; Fuji systems, Tokyo, Japan) was used during grafting. On pump isolated CABG was performed utilizing the same techniques. All on pump CABG was performed while the heart was beating. Grafting was always performed from the left internal mammary artery to the left anterior descending coronary artery, followed by grafting of the circumflex coronary artery and right coronary artery using a radial artery or a saphenous vein. The bilateral internal mammary was used in the non-touch aorta technique and ascending aortic sclerosis or calcification was assessed based on pre-operative findings from imaging examinations and intra-operative palpation. The quality of the anastomosis was assessed post-graft utilizing a transit-time flow probe (VeriQ System and TTFM probes; Medistim ASA, Oslo, Norway).

The primary endpoint was postoperative 30-day mortality (death occurring within 30 days after surgery). Secondary endpoints included major postoperative complications, such as low cardiac output syndrome, myocardial infarction, bleeding requiring surgical intervention, stroke, postoperative atrial fibrillation, mediastinitis, intubation time longer than 72 hours, ICU stay longer than 7 days, and postoperative length of stay longer than 30 days. Occurrence

of any short runs of atrial fibrillation more than 30 seconds during the hospital stay was considered to represent an occurrence atrial fibrillation. Major adverse cardiac and cerebrovascular events (MACCE) included death, acute myocardial infarction, cerebrovascular event, or further revascularization by percutaneous coronary intervention or CABG. The neurologic event was defined as an endpoint when symptoms appeared and could be corroborated using computed tomography (CT) and magnetic resonance imaging (MRI). The final diagnosis was performed by a neurosurgeon and it was considered a neurologic event if diagnosed. If there were no visual findings, the transient ischemic attack was not included.

## Statistical analysis

Continuous variables were expressed as the mean and standard deviations or the median and interquartile ranges, and categorical variables were shown as frequencies or percentages. Continuous data were analyzed using the Independent Student's t-test or Mann-Whitney U-test. Categorical variables were analyzed by Chi-Squares and Fischer's exact test. The in-hospital survival rates and MACCE-free rates after surgery between two groups were determined by Kaplan-Meier survival curves and compared by the log-rank test. Analyses were conducted with JMP software, version 10 (SAS Institute Japan, Tokyo, Japan).

## Results

A total of 393 consecutive patients were included in this study, with 85 patients (group A, 22%) considered high-risk and placed in the prophylactic IABP group and 308 patients (group B, 78%) considered non-high risk patients not allocated to prophylactic IABP. The preoperative clinical data are listed in **Table 1**. There were no significant differences in age, gender, body mass index, comorbidities and insulin use between both groups. In patients in the group A, preoperative low ejection fraction (group A versus group B: 50 ± 18 vs 54 ± 14%, p<0.0001), emergency operation (19 [22.4%] vs 24[7.8%], respectively; p = 0.001), NYHA III/ IV functional class (30 [35.3%] vs 74 [24.1%], respectively; p = 0.051, left main coronary stenosis (45 [53%] vs 105 [37%], respectively; p = 0.009), and the current smoker (21 [24.6%] vs 29 [9.4%], respectively; p = 0.003) were more frequent. For recent AMI, the progress of all members was confirmed and the results were listed in Table 1. A recent AMI was defined as being diagnosed with AMI when hospitalized. There were no differences between the two groups (12 [14.1%] vs 31 [10.1%], respectively; p = 0.326). Our policy is that if there is a time allowance for those who have surgical indications for carotid artery lesions, they will be treated first, but in this study there were no such patients. As a treatment for stenotic lesion of the carotid artery and its treatment, none of the patients in this patient group underwent a surgical operation (including stent insertion). Nine patients had pointed stenosis of the carotid artery before surgery and all were consulted for neurosurgery at this hospital. Additionally, single photon emission computed tomography was performed. It was decided that there was little need for preoperative surgical treatment.

Extensive rehabilitation had been performed in 19 (4.8%) patients. There was a significant difference between the prophylactic IABP group that underwent extensive rehabilitation (8 [9.4%] vs 11 [3.6%], respectively; p = 0.041) (Table 2). Seventy-nine percent (15 patients) of the patients started with in-bed exercise and eventually advanced to out-of-bed exercise (63% [5 of 8] vs 91% [10 of 11], respectively; p = 0.262). Reasons for extensive rehabilitation (the reason why ADL decreased) were due to many factors including the onset of heart failure (8 patients), the onset of AMI (6 patients), temporarily use of IABP after hospitalization (3 patients), onset of AMI and heart failure (1 patient), and brain infarction (1 patient). In this

**Table 2. Preoperative extensive rehabilitation.**

| Result | Total (N = 393) | Prophylactic IABP (n = 85) | No prophylactic IABP (n = 308) | p-value (prophylactic IABP vs. non-prophylactic) |
|---|---|---|---|---|
| **Extensive rehabilitation, %** | 4.8 (19 of 393) | 9.4 (8 of 85) | 3.6 (11 of 308) | 0,041 |
| **Rehabilitation period, days** | | | | 0.065 |
| Mean ± SD | 25 ± 13 | 19 ± 11 | 30 ± 13 | |
| Median (IQR) | 23.00 [16.00, 37.00] | 19.5 [9.25, 22.75] | 36.00 [19.00, 38.00] | |
| **In-bed exercises, %** | 100 (19 of 19) | 100 (8 of 8) | 100 (11 of 11) | 1 |
| **Out-of-bed exercise, %** | 79 (15 of 19) | 63 (5 of 8) | 91 (10 of 11) | 0.262 |
| **Exit ICU, %** | 79 (15 of 19) | 63 (5 of 8) | 91 (10 of 11) | 0.262 |

SD: Standard Deviation, IQR: Interquartile Range, ICU: Intensive Care Unit, in-bed exercises = Achievement of in-bed exercises, out-of-bed exercise = Achievement of out-of-bed exercise

study, only one patient underwent surgery with the need to halt extensive rehabilitation (Table 2).

At the induction of anesthesia and just before the start of surgery, group B tended to have higher blood pressure and less use of catecholamines, and group A had higher cardiac index (Table 3).

Intra and postoperative results are shown in **Table 4**. There were no significant differences in the operation time, using cardiopulmonary bypass, pump time, converted to on-pump CABG, reoperation for bleeding, required transfusion of red blood cells, leg wound problems, leg wound infection, occurrence of mediastinitis and neurologic events. There was a 93% post-operative follow-up rate over an average of 42.5 months (± 42.6). The follow-up results after discharge were confirmed on the medical records of our hospital, related hospitals, or via telephone. The mean number of distal anastomoses were few in group B patients (2.8 ± 1.1 vs 2.3 ± 1.0; p<0.0001). Post-operative occurrence of AKI between the two groups (3 [3.6%] vs 6 [2.1%], respectively; p = 0.326). However, patients without prophylactic IABP required more dialysis following cardiac surgery (0 of 3 [0%] vs 6 of 7 [86%], respectively; p = 0.033). Only one patient (without prophylactic IABP) with end-stage renal disease requiring maintenance hemodialysis therapy initiated chronic dialysis following cardiac surgery. The duration of mechanical ventilation (1.4 ± 1.4 vs 1.1 ± 1.6 days; p = 0.127), the length of ICU stay (4.2 ± 4.4 vs 4.9 ± 2.3 days; p = 0.181) and the length of hospital stay (25 ± 18 vs 22 ± 15 days; p = 0.095) were no difference between the two groups ("ICU stay" was not counted as ICU admission stay during the preoperative period.), yet significantly difference was diagnosed with the prolonged ventilation>72 hours (11 [12.5%] vs 13 [4.2%], respectively; p = 0.014).

IABP was removed postoperatively after extubation if the patient was hemodynamically stable. No IABP-related complications were reported and there were no instances of IABP-related mortality. Forty-four patients in the non-high-risk group (14%) required IABP support perioperatively, but are not included in the original high-risk prophylactic IABP group.

Overall inpatient surgical mortality was 1.3% (5/393) secondary to postoperative low cardiac output syndrome (n = 3), sepsis (n = 1), and multiple organ failure (n = 1). Mortality in the prophylactic IABP group was 0% (0/85).

Overall 30-day mortality was 0% versus 1.0% in the prophylactic IABP group. Major adverse cardiac or cerebrovascular events (MACCE) occurred in 3.8% overall (including after discharge), with a 5.9% MACCE in the prophylactic IABP group. There were no significant difference in 30 days mortality (group A vs group B = 0 [0%] vs 3 [1%], respectively; p = 1.0) as well as in-hospital deaths (0 [0%] vs 5 [1.6%], respectively; p = 0.589). The 12-months survival

**Table 3. Clinical outcomes Pre- and Postoperative data with or without prophylactic IABP patients.**

| Result | Total (N = 393) | Prophylactic IABP (n = 85) | No prophylactic IABP (n = 308) | p-value (comparing prophylactic IABP and non-prophylactic) |
|---|---|---|---|---|
| **Preoperative (at the induction of anesthesia)** | | | | |
| shock state—Yes, % | 2 | 1.2 | 2.3 | 1 |
| ECMO use, % | 0 | 0 | 0 | |
| Systolic BP—mmHg, Mean (SD) | 131 ± 24 | 125 ± 22 | 133 ± 25 | 0.113 |
| Mean BP—mmHg, Mean (SD) | 88 ± 15 | 82 ± 13 | 89 ± 15 | 0.0001 |
| CI L/min/m2 | 2.5 ± 0.7 | 2.8 ± 0.7 | 2.4 ± 0.7 | 0.008 |
| Catecholamines; n/ total (%) | 10.4 | 31.8 | 4.6 | <0.0001 |
| Dosage (μg/kg/min)—mean SD | | | | |
| Dopamine | 0.15 ± 0.77 | 0.33 ± 0.78 | 0.10 ± 0.77 | 0.018 |
| Dobutamine | 0.10 ± 0.53 | 0.27 ± 0.76 | 0.05 ± 0.43 | 0.0005 |
| Norepinephrine | 0.0004 ± 0.0064 | 0 ± 0 | 0.0005 ± 0.0072 | 0.529 |
| Epinephrine | 0 ± 0 | 0 ± 0 | 0 ± 0 | |
| CI (at the start of operation) L/min/m2 | 2.5 ± 0.6 | 2.7 ± 0.7 | 2.4 ± 0.6 | 0.0006 |
| **Postoperative (before admission in ICU)** | | | | |
| shock state—Yes, % | 0.51 | 0 | 0.65 | 0.459 |
| ECMO use, % | 0.51 | 0 | 0.65 | 0.459 |
| Systolic BP—mmHg, Mean (SD) | 101 ± 16 | 99 ± 14 | 102 ± 17 | 0.113 |
| Mean BP—mmHg, Mean (SD) | 67 ± 10 | 64 ± 10 | 67 ± 11 | 0.031 |
| CI L/min/m2 | 2.8 ± 0.7 | 2.9 ± 0.8 | 2.8 ± 0.7 | 0.551 |
| Catecholamines; n/ total (%) | 98.2 | 98.8 | 98.1 | 1 |
| Dosage (μg/kg/min)—mean SD | | | | |
| Dopamine | 2.16 ± 1.36 | 2.16 ± 1.09 | 2.16 ± 1.42 | 0.989 |
| Dobutamine | 0.34 ± 0.98 | 0.39 ± 0.92 | 0.34 ± 1.00 | 0.655 |
| Norepinephrine | 0.04 ± 0.43 | 0.08 ± 0.66 | 0.03 ± 0.34 | 0.344 |
| Epinephrine | 0 ± 0 | 0 ± 0 | 0 ± 0 | |

ECMO: Extracorporeal membrane oxygenation, BP: blood pressure, SD: Standard Deviation, CI: Cardiac index, ICU: Intensive Care Unit

curves in the 85 high-risk patients who received prophylactic IABP compared to the 308 non-high risk patients who did not receive prophylactic IABP were similar (p = 0.846) (Fig 2). The postoperative MACCE free rates (1 year) was 69% (group A) vs 69%(groupB) (p = 0.970) (Fig 3).

## Discussion

Currently, the consensus on the benefit of prophylactic IABP is not so widespread. Some papers were reported the Prophylactic IABP insertion in high-risk patients undergoing cardio-thoracic surgery has been shown to reduce postoperative mortality [2, 6–8]. *Dyub* et al. reported that patients who received prophylactic IABP preoperatively had a mortality benefit with an odds ratio (OR) of 0.41 (95% CI 0.21–0.82; p = 0.01) [6]. Multiple studies reported the benefit of prophylactic IABP [1,2, 6–8], but it had not been established as the gold standard for high-risk CABG. One of the reasons was complications. Patients undergoing IABP insertion, however, have been considered at higher risk for rebleeding, and prolonged ventilation, as well as at increased risk of reintubation, tracheostomy, dialysis, and prolonged ICU stay [9, 10].

**Table 4. Clinical outcomes and complications with or without prophylactic IABP patients.**

| Result | Total (N = 393) | Prophylactic IABP (n = 85) | No prophylactic IABP (n = 308) | p-value (prophylactic IABP vs. non-prophylactic) |
|---|---|---|---|---|
| Follow up, % | 93 | 93 | 94 | 0.808 |
| Observation period, months | | | | 0.037 |
| Mean ± SD | 42.5 ± 42.6 | 33.9 ± 34.1 | 44.9 ± 44.6 | |
| Median (IQR) | 27 [5, 73] | 23 [5.5, 51] | 28 [5, 82] | |
| Operation time, min, Median (IQR) | 258.00 [210.00, 313.00] | 256.00 [206.50, 304.00] | 259.50 [211.25, 314.00] | 0.466 |
| Off-Pump CABG | 56.7 | 50.6 | 58.4 | 0.217 |
| Pump time, min, Median (IQR) | 118.00 [96.00, 148.00] | 109.00 [93.00, 144.00] | 120.00 [101.00, 149.75] | 0.217 |
| converted to on-pump CABG, % | 2.8 | 2.4 | 2.9 | 1.000 |
| Postoperative Max SCr, mg/dl, Mean ± SD | 2.3 ± 9.7 | 1.0 ± 0.4 | 2.9 ± 11.8 | 0.383 |
| AKI, % | 2.4 | 3.6 | 2.1 | 0.423 |
| required Dialysis, % | 60 | 0.0 | 86 | 0.033 |
| reoperation for bleeding, % | 1.3 | 1.2 | 1.3 | 1.000 |
| Distal anastomoses | | | | <0.001 |
| Mean ± SD | 2.5 ± 1.0 | 2.8 ± 1.0 | 2.4 ± 0.9 | |
| Median (IQR) | 2.00 [2.00, 3.00] | 3.00 [2.00, 3.00] | 2.00 [2.00, 3.00] | |
| required transfusion of red blood cells, % | 59 | 68.2 | 57 | 0.062 |
| Leg wound problems, % | 0.7 | 0.0 | 1 | 1.000 |
| Leg wound infection, % | 0.3 | 0.0 | 0.3 | 1.000 |
| Neurologic events, % | 0 | 0 | 0 | - |
| Duration of IABP support, days | | | | 0.040 |
| Mean ± SD | 2 ± 1.6 | 1.8 ± 1.2 | 2.4 ± 2.1 | |
| Median (IQR) | 1.50 [1.00,2.00] | 1.00 [1.00,2.00] | 2.00 [1.00,3.00] | |
| Duration of mechanical ventilation (post operative days) | | | | 0.127 |
| Mean ± SD | 1.2 ± 1.6 | 1.4 ± 1.4 | 1.1 ± 1.6 | |
| Median (IQR) | 1.00 [1.00, 1.00] | 1.00 [1.00, 1.00] | 1.00 [1.00, 1.00] | |
| Prolonged ventilation >72 hours, % | 6.1 | 12.5 | 4.2 | 0.014 |
| Reintubation, % | 1.4 | 0 | 1.8 | 0.590 |
| ICU stay (post operative days) | | | | 0.181 |
| Mean ± SD | 4.3 ± 4.0 | 4.2 ± 4.4 | 4.9 ± 2.3 | |
| Median (IQR) | 3.00 [3.00, 4.00] | 4.00 [3.00, 6.00] | 3.00 [3.00, 4.00] | |
| ICU stay longer than 7 days, % | 6.6 | 11.4 | 5.3 | 0.071 |
| Length of hospital stay.days | | | | 0.095 |
| Mean ± SD | 23 ± 16 | 25 ± 18 | 22 ± 15 | |
| Median (IQR) | 19.00 [16.00, 24.00] | 20.00 [17.00, 25.00] | 19.00 [16.00, 23.00] | |
| postoperative length of stay longer than 30 days, % | 12.8 | 14.8 | 12.3 | 0.573 |
| Post operative atrial fibrillation,% | 11.9 | 13.4 | 11.4 | 0.699 |
| Mediastinitis, % | 1.6 | 1.2 | 1.7 | 1.000 |
| 30 days mortatlity, % | 0.8 | 0.0 | 1 | 1.000 |
| In-hospital deaths, % | 1.3 | 0.0 | 1.6 | 0.589 |
| MACCE, % | 3.8 | 5.9 | 3.3 | 0.333 |

IQR: Interquartile Range, SD: Standard Deviation, CABG: Coronary Artery Bypass Grafting, SCr: Serum Creatinine, ICU: Intensive Care Unit, MACCE: Major Adverse Cardiac and Cerebrovascular Events

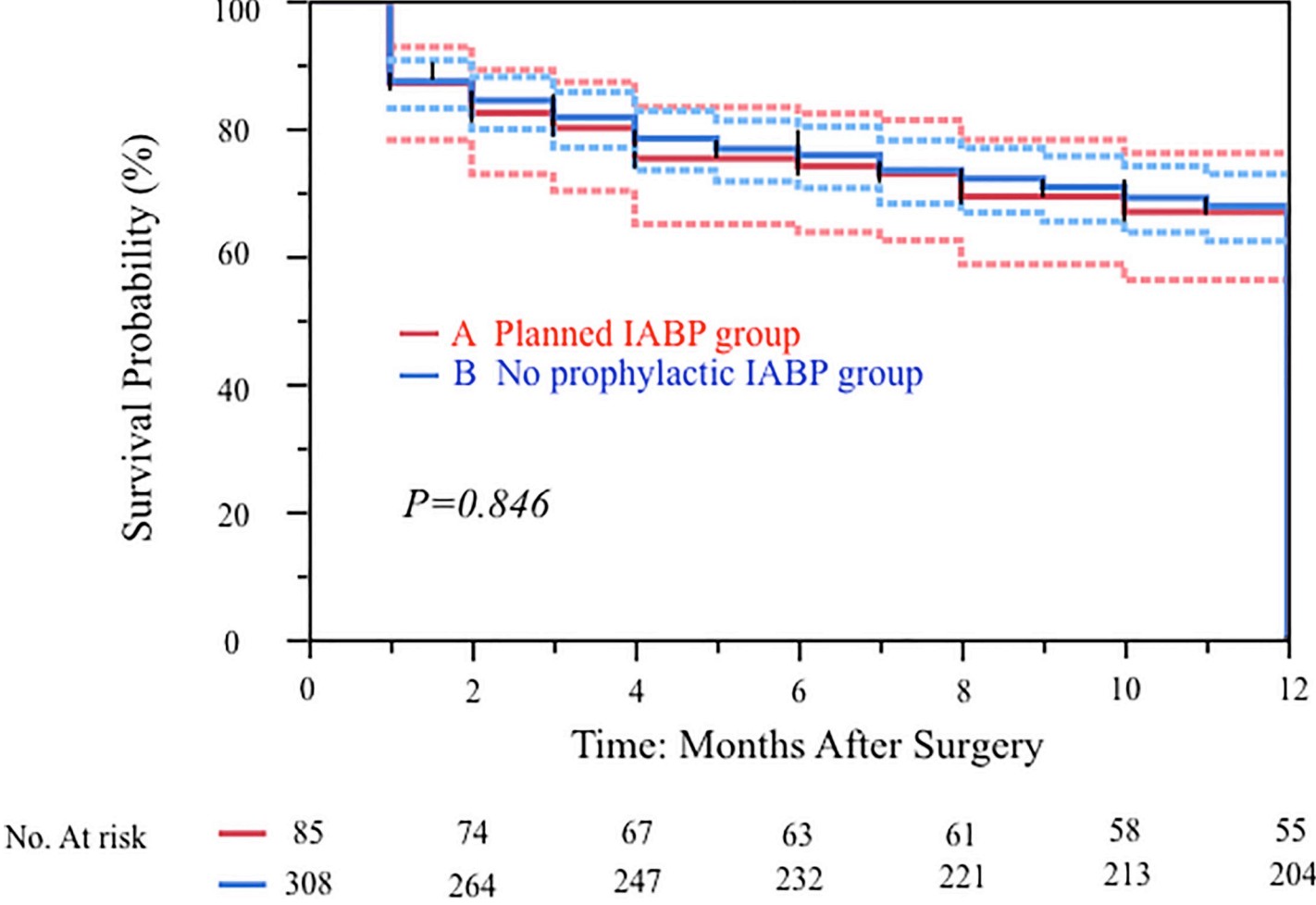

**Fig 2. Kaplan-Meier curves for freedom from overall survival of 393 patients with isolated CABG in our institution: 85 patients with prophylactic IABP group and 308 patients with non-prophylactic IABP patients.**

Yet, when patients are properly evaluated preoperatively for possible contraindications to IABP (such as peripheral vascular disease) and strict postoperative surveillance is maintained, the risk of IABP-related complication can be minimized. In our studies, the success rate of IABP placement was 100% in cases where IABP placement was planned. For all IABP placements, we utilized a percutaneous approach, advancing through the femoral artery. Prior to surgery, we performed CT imaging to ensure there were no problems with the access of the patient.

Nevertheless, while the conclusion of previous studies points towards a benefit with preoperative IABP, the results have been controversial due to lack of both prophylactic IABP insertion criteria and lack of definition of what constitutes a high-risk patient. In our study, we aimed to avoid these limitations by using the definition of a high-risk patient as illustrated by *Ding* et al [11]. Furthermore, the criteria for IABP insertion in our study cohort was strictly prophylactic, rather than therapeutic such as in low cardiac output syndrome, similar to the methods utilized by *Shi* et al [12].

Although we were unable to determine the advantages of using prophylactic IABP, some papers report good results.A recent meta-analysis (MTA) reported that prophylactic IABP use

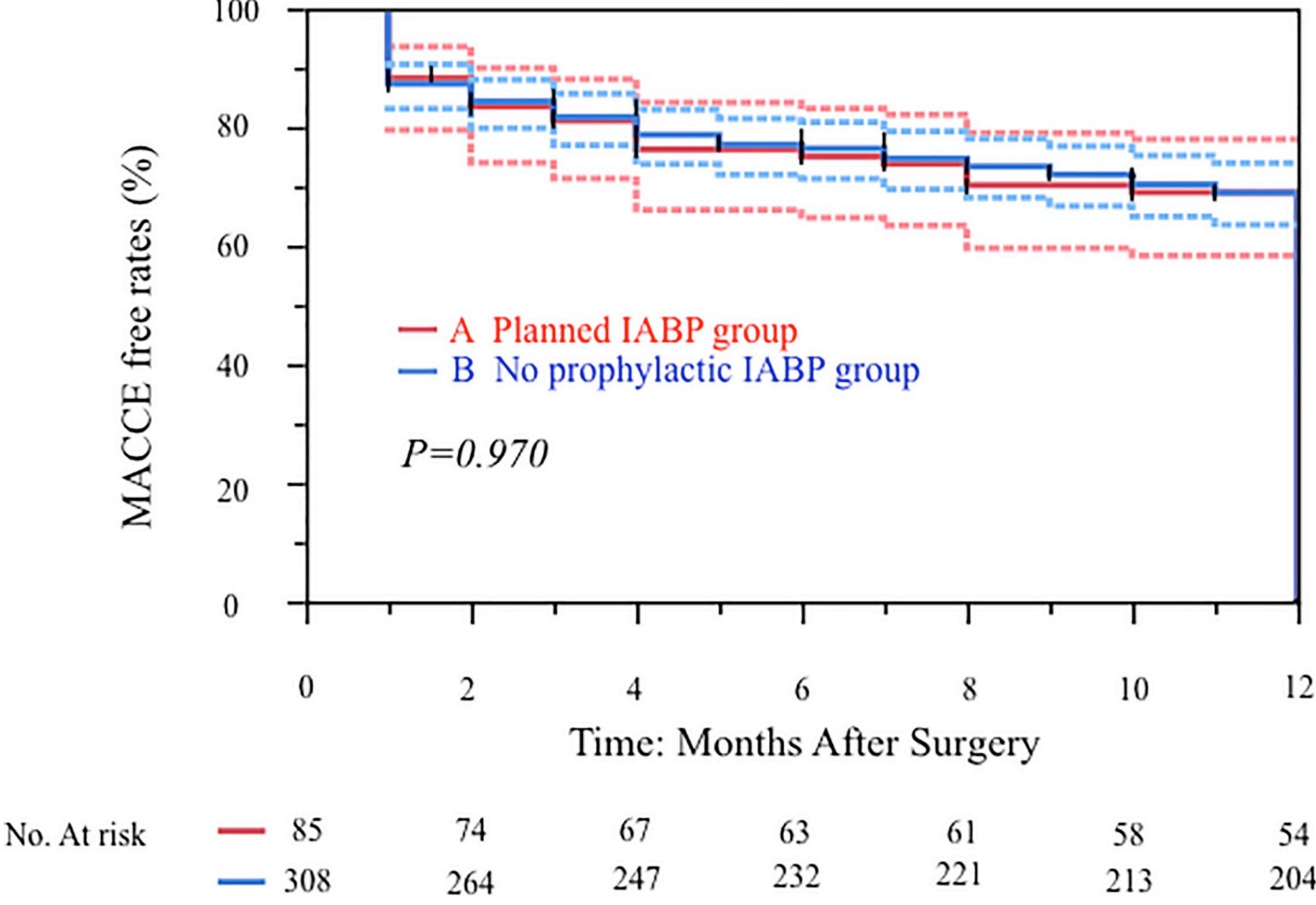

**Fig 3. Kaplan-Meier curves for MACCE-free rates of 393 patients with isolated CABG in our institution: 85 patients with prophylactic IABP group and 308 patients with non-prophylactic IABP patients.**

reduced mortality and MACCE in high-risk CABG patients [13]. *Poirier* et al showed that hospital death following preoperative IABP was 2.5%, compared to 12.6% in the control (non-IABP) group. Moreover, from the observational study, the hospital death following preoperative IABP use was 7.6%, compared to 2.4% in the non-use group [14]. In our report, the hospital death in the preoperative IABP group was 0%, compared to 1.6% in the non-use group; there were no differences between the two groups. The significance of IABP is inconsistent based on findings from different studies. In our studies, the preoperative IABP group was a high-risk group; therefore, the use of preoperative IABP may have had an advantage in improving the survival rate. However, future studies are warranted to definitively elucidate the significance of IABP on patient outcomes. *Jiayang* et al showed that preoperative prophylactic IABP reduced the incidence of CABG-associated AKI[15]. Our results also showed that postoperative dialysis was significantly less in the prophylactic IABP group (0 of 3 [0%] vs 6 of 7 [86%], respectively; p = 0.033). The utility of prophylactic IABP is still open for further study, but early studies indicate that postoperative results may be improved. However, further research is warranted to demonstrate the beneficial effects of prophylactic IABP use.

Our study illustrates the advantages of prophylactic IABP are manifold, including ease of safe placement through fluoroscopy-guidance, improved circulation support at the induction of anesthesia, and circulatory support in the setting of possible postoperative low cardiac output syndrome. Nevertheless, this study reported that no survival advantage was found in the prophylactic application of an IABP in hemodynamically stable we had not experienced IABP-related complications.

Compared to average values, we observed better outcomes in regard to 30-day mortality (group A vs group B = 0 [0%] vs 3 [1%]) and in-hospital deaths (0 [0%] vs 5 [1.6%]). First, a multidisciplinary team of cardiologists, cardiac surgeons, nurses, rehabilitation staff, and medical engineers met to outline a detailed treatment plan for patients, which helped avoid preoperative risks and increase survival rates. Second, preoperative extensive rehabilitation might have been something related to prevent occurring perioperative complication, Third, postoperatively, we continued hospitalization and monitored patient progress using a coronary angiogram and echocardiography. There was a tendency for postoperative hospitalization days to be prolonged, but patients were fully rehabilitated and discharged in a stable state.

Compared to a study conducted by *Poirier* et al [14], we found that our prophylactic IABP group tended to have longer ICU stay times (MTA vs. our results: mean 1–4 days vs 4.3 days). However, in their study, *Poirier* et al stated that "it is important to note that the rate of IABP cross-over (IABP installed during or after cardiac surgery) varied from 13% to 100%", which is a drastic fluctuation. On the other hand, the timing of IABP removal drastically varies from one study to another; *Poirier* suggests that these inconsistencies are due to the lack of data on the optimal duration for IABP use. At our institution, IABP removal was performed following hemodynamic stability, which resulted in longer IABP placement periods (IABP cross-over). This seemed to be the cause of the prolonged ICU length-of-stay in the prophylactic IABP group in our studies compared to the recently published MTA. Similarly, for prolonged intubation times post-surgery, heart failure was controlled, circulation was stabilized, and then intubation was performed. The need to control heart failure and stabilize circulation was more prevalent in the high risk group, which could justify the prolonged intubation times following surgery. These are the treatment strategies employed by our institution, which seemed to yield results that contradict findings from recently published studies on IABP. One of the reasons for the low incidence of neurologic events was extensive screening prior to surgery utilizing head and neck CT, MRI and carotid artery echocardiography. Cases deemed necessary might receive treatment for cerebrovascular disease prior to cardiac surgery. We also used epiaortic echocardiography during surgery when it was deemed necessary. In patients with atrial fibrillation, we administered heparin following surgery. Additionally, defibrillation after onset of atrial fibrillation was performed immediately. Although there is no known correlation between our treatment approach and the incidence of neurological events, the occurrence of neurologic events was not evident in this study.

We acknowledge the limitations of this study, such as the lack of a randomized control trial to allow for fixed evaluation of prophylactic IABP during CABG, the limited generalizability secondary to being conducted at a single center, as well as its small cohort size. The nonrandomized design might have affected our results, owing to unmeasured confounds, procedural bias, or detection bias. However, we believe that our study allows a real-life evaluation of prophylactic IABP, particularly its importance at the level of a community hospital, such as ours. We used the definition of high-risk patient based on *Ding* et al's report. That was the definition at the time of OPCAB, and that definition may not be applicable to our patient selection in this study. The use of preoperative IABP may involve circulatory dynamics during anesthesia induction, during the weaning of cardiopulmonary bypass, and management immediately after weaning. It may also be involved in stabilizing the circulatory dynamics of patients in the

acute phase after surgery. However, these are all speculations and could not be stipulated from this study.

This study has several limitations. First, the number of patients was relatively small. Second, the study was performed at a single center; therefore, the results might not be generalizable to other centers in different situations. The nonrandomized design might have affected our results, owing to unmeasured confounds, procedural bias, or detection bias.

## Conclusion

Given the lack of IABP-related complications in this study cohort, our prospective observational study shows that prophylactic IABP in the high-risk patient undergoing CABG is an acceptable option.

## Supporting information

**S1 File. Observation data.**
(XLSX)

## Acknowledgments

We thank JAM Post (www.jamp.com) for English language editing.

## Author Contributions

**Conceptualization:** Ken Nakamura, Cholsu Kim, Hideaki Uchino, Takao Shimanuki.

**Data curation:** Ken Nakamura, Kimihiro Kobayashi.

**Formal analysis:** Ken Nakamura, Ri Sho.

**Investigation:** Azumi Hamasaki, Tetsuro Uchida.

**Methodology:** Azumi Hamasaki, Tetsuro Uchida, Mitsuaki Sadahiro.

**Project administration:** Ken Nakamura, Azumi Hamasaki.

**Resources:** Ken Nakamura.

**Supervision:** Mitsuaki Sadahiro.

**Writing – original draft:** Ken Nakamura, Azumi Hamasaki.

**Writing – review & editing:** Ken Nakamura, Azumi Hamasaki, Tetsuro Uchida, Mitsuaki Sadahiro.

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
