## [Decision Letter · Decision Letter 0]

27 Aug 2019

PONE-D-19-19063

The use of prophylactic intra-aortic balloon pump in high-risk patients undergoing Coronary Artery Bypass Grafting

PLOS ONE

Dear Dr Nakmaura,

Thank you for submitting your manuscript to PLOS ONE. After careful consideration, we feel that it has merit but does not fully meet PLOS ONE’s publication criteria as it currently stands. Therefore, we invite you to submit a revised version of the manuscript that addresses the points raised during the review process.

We would appreciate receiving your revised manuscript by Oct 11 2019 11:59PM. To enhance the reproducibility of your results, we recommend that if applicable you deposit your laboratory protocols in protocols.io, where a protocol can be assigned its own identifier (DOI) such that it can be cited independently in the future. For instructions see: http://journals.plos.org/plosone/s/submission-guidelines#loc-laboratory-protocols

We look forward to receiving your revised manuscript.

Kind regards,

Andrea Ballotta

Academic Editor

PLOS ONE

**Journal Requirements:**

2. Thank you for stating that “The funders had no role in study design, data collection and analysis, decision to publish, or preparation of the manuscript” in your financial disclosure.

Please also provide the name of the funders of this study (as well as grant numbers if available) in your financial disclosure statement.

**Additional Editor Comments (if provided):**

Thanks for your contribution but, as stated by the reviewers, the manuscript needs major revision.

**Comments to the Author**

1. Is the manuscript technically sound, and do the data support the conclusions?

Reviewer #1: Partly

Reviewer #2: Yes

2. Has the statistical analysis been performed appropriately and rigorously? 

Reviewer #1: No

Reviewer #2: Yes

3. Have the authors made all data underlying the findings in their manuscript fully available?

Reviewer #1: Yes

Reviewer #2: Yes

4. Is the manuscript presented in an intelligible fashion and written in standard English?

Reviewer #1: No

Reviewer #2: Yes

5. Review Comments to the Author

Reviewer #1: The article presents a prospective observational analysis of 99 patients (group A; 21%) at “high-risk” (out of 471) having undergone from Decembre 2005 to December 2017 either isolated or combined CABG with prophylactic IABP compared to others (Group B 79%, 372 pts). All patients with contraindications to IABP were excluded from the study. The “high risk” criteria the patient should have a stable hemodynamics with 2 criteria out of 6 (NYHA III or IV; LVEF<40%; LVEDD>65mm; LM stenosis >50%; a diffuse CAD defined as more than three distal or more anastomosis and refractory unstable angina.

IABP were inserted the day before operation in CathLab from femoral artery and with a subsequent pre and postoperatively ICU monitoring.

Primary endpoint: 30D mortality

Secondary endpoint: major postoperative complications (LCO, postoperative MI; bleeding; stroke; postoperative AF; IOT>72hh; ICU stay >7dd; total postoperative days >30 dd and MACCE.

Forty seven pts in Group B required IABP postoperatively.

Mean follow up was 32 months.

The two groups were significantly different for LVEF, NYHA class, LM disease; emergent operation and number of stenosis but similar for age, sex, BMI and other comorbidities. Hemodialysis and current smoker were higher in Group B.

The results show similar intraoperative results (time, conversion, postop bleeding, transfusion and strokes).

In the postoperative results Group A had a longer mechanical ventilation and ICU stay.

No complications are reported for the support of IABP.

Overall impatient mortality was 2.1% (2% in group A) (30d 2% Group A vs 1.4% Group B) and no differences in terms of death and MACCE were found.

In the discussion section the authors explain their clinical results and their strategies:

- a multidisciplinary team of cardiologists, cardiac surgeons, nurses, rehabilitation staff, and medical engineers met to outline a detailed treatment plan for patients

- all non-urgent patients underwent extensive cardiac rehabilitation prior to surgery, even patients that were admitted into the emergency room

- all patients underwent treated dental conditions, glycemic imbalances, and co-morbid treatable diseases such as carotid artery stenosis prior to cardiac surgery.

- post-operatively a prolonged hospitalization were used to monitor patient’s progress using a coronary angiogram and echocardiography.

-

The authors, in the end, stated that prophylactic IABP in the high-risk patient undergoing CABG is an acceptable option

Even if the article is interesting, the results are unremarkable and is a hot topic, in my opinion some issues must be solved and more details must be added.

- Patients selection is the key of this article and in my opinion the authors must evaluate just isolated CABG and not combinated ones. Furthermore is not clear the types of operations in each group.

- Surgical description is very poor and there is no mention about conduits and surgical techniques.

- In my opinion the evaluation of the “high risk” patients is not correct in this population. The use of the definition of Ding et al could be useful only in OPCABG because includes technical aspects of off pump surgery, beyond the clinical features, that is inconsistent during ONCABG. Less than 50% of patients underwent OPCABG in your population.

- Your clinical strategy leads to a strict preoperative selection and preparation of patients and even if it is probably very far from my clinical practice I would like to better understand it. So from the first access to the hospital how long do you take for the “exstensive rehabilitation” and which are your endpoints? this strategy also adapt to patients with left main stenosis? You didn’t clarify how many AMI (recent of previous) was present in your population. Probably another interesting evaluation could be made on your strategy for carotid stenosis treatment: how many patients received a surgical or endovascular treatment of the carotid arteries in your population and how long before?

- What about postoperative AKI?

- There is no mention about conduction and completeness of FU.

- In my opinion an actuarial survival estimated by KM analysis has very little sense in a period of observation of 30 days. It should be made on the first year at least.

- The topic is hot and the consensus on the benefit of prophylactic IABP is not so widespread. So in my opinion could be important to emphasize the open question in the discussion section and the references must be enlarged. Furthermore the recent article from Rampersad et al is not a RCT but a metaanalysis.

- The tables are very difficult to read, please reorganize and simplify

Reviewer #2: Thank you for your paper.

It is a very interesting field because IABP seems has to be abandoned by cardiological guidelines but still it has been use consistently in cardiac surgery

As you underlined the cohort of patients in not very wide and is not possible to be definitive on the prophylactic IABP with this numbers

Where do you allocate the patient in the preoperative period? has to be at least an HDU. Has this time counted in the ICU admission stay? In my institution we insert the IABP after the induction in OR.

It seems that the method you have used to detect high risk patients is not working properly since there is a consistent use in not high risk group.

Plus is very interesting that the high risk LVEF, despite is significantly different from the other group, is 49%.

The last consideration is on the weaning time. Have you got any standard parameter to consider to asses the possibility to wean? Do you you use levosimendan?

6. PLOS authors have the option to publish the peer review history of their article (what does this mean?). If published, this will include your full peer review and any attached files.

Reviewer #1: No

Reviewer #2: No

---

## [Author Response · Author response to Decision Letter 0]

28 Sep 2019

Date: September 26, 2019

Andrea Ballotta

Academic Editor

PLOS ONE

Dear Dr. Andrea:

We appreciate the reviewer’s insightful comments on our manuscript entitled, “The use of prophylactic intra-aortic balloon pump in high-risk patients undergoing Coronary Artery Bypass Grafting”. We have revised the manuscript to address the reviewers’ concerns and have included our responses to the reviewers’ comments below.

Thank you for your consideration. I look forward to hearing from you again.

Sincerely,

Ken Nakamura, MD

Division of Cardiovascular Surgery, 

Nihonkai General Hospital, 30 Akihochou, 

Sakata, 998-8501, Japan.

Tel: +81-23-628-5342

Fax: +81-23-628-5345

E-mail: ken.nakamura622@gmail.com

Reviewer #1

1 Comment: Patients selection is the key of this article and in my opinion the authors must evaluate just isolated CABG and not combinated ones. Furthermore is not clear the types of operations in each group.

Response:

We thank the reviewer for these insightful comments. In accordance with this comment, we evaluated isolated CABG, rather than combined ones. All the obtained values were listed in tables 1, 3, and 4. Three hundred ninety-three patients were included in the isolated CABG group. All statistical analyses were repeated. All applicable locations and numbers have been changed. We added the following sentence to the Patients and Methods section: “at Nihonkai General Hospital, and involved 471 unique patients from December 2005 to December 2017 who underwent isolated or combined CABG. Of those, 393 patients underwent isolated CABG and were included for the analysis.” (Page 5; line 51)

2 Comment: Surgical description is very poor and there is no mention about conduits and surgical techniques.

Response: 

Thank you for your comment. In accordance with this comment, we have expanded our Patients and Methods section to include the following, “A CPB circuit was used when deemed necessary during the preoperative assessment. It primarily included criteria such as a large left ventricle and low cardiac function. Additionally, on-pump CABG was scheduled following a comprehensive risk assessment, which included the location and quality of target vessels, and accounted for technically challenging cases. If complete revascularization was feasible, OPCAB was scheduled. Conversion to CPB was considered if there was any evidence of hemodynamic instability, such as ventricular arrhythmia, hypotension (systolic pressure ≤ 80 mmHg), and cardiac arrest during OPCAB procedures. The OPCAB is performed after a median sternotomy. The heart was displaced using a posterior pericardial stitch, gauze, and a tissue stabilizer (Octopus Evolution tissue stabilizer and Octopus Evolution AS tissue stabilizer; Medtronic Corporation, Minneapolis, MN, USA). Body position changes and gravity support (Trendelenburg, right and left table rotations) were carried during surgery. A CO2 blower mister device was used in situations where a bloodless field was not achieved after proximal target vessel occlusion. An intracoronary shunt (Phycon coronary minishunt; Fuji systems, Tokyo, Japan) was used during grafting. On pump isolated CABG was performed utilizing the same techniques. All on pump CABG was performed while the heart was beating. Grafting was always performed from the left internal mammary artery to the left anterior descending coronary artery, followed by grafting of the circumflex coronary artery and right coronary artery using a radial artery or a saphenous vein. The bilateral internal mammary was used in the non-touch aorta technique and ascending aortic sclerosis or calcification was assessed based on pre-operative findings from imaging examinations and intra-operative palpation. The quality of the anastomosis was assessed post-graft utilizing a transit-time flow probe (VeriQ System and TTFM probes; Medistim ASA, Oslo, Norway).” (Page 9; line 102).

3 Comment: In my opinion the evaluation of the “high risk” patients is not correct in this population. The use of the definition of Ding et al could be useful only in OPCABG because includes technical aspects of off pump surgery, beyond the clinical features, that is inconsistent during ONCABG. Less than 50% of patients underwent OPCABG in your population.

Response: Thank you for your insightful comment. As stated, the definition outlined by Ding et al. was the definition at the time of OPCAB, and may not be applicable to our patient selection in this study. I will add it to the Limitation as follows “We used the definition of high-risk patient based on Ding et al’s report. That was the definition at the time of OPCAB, and that definition may not be applicable to our patient selection in this study.” (Page 19; line 298)

There was no conclusion that the use of IABP was effective with On Pump CABG, and I thought there were some opinions that it was not necessary. We added the following sentence in the Discussion section: “The use of preoperative IABP may involve circulatory dynamics during anesthesia induction, during the weaning of cardiopulmonary bypass, and management immediately after weaning. It may also be involved in stabilizing the circulatory dynamics of patients in the acute phase after surgery. However, these are all speculations and could not be stipulated from this study.”

 (Page 19; line 300).

4 Comment: Your clinical strategy leads to a strict preoperative selection and preparation of patients and even if it is probably very far from my clinical practice I would like to better understand it. So from the first access to the hospital how long do you take for the “exstensive rehabilitation” and which are your endpoints? this strategy also adapt to patients with left main stenosis? You didn’t clarify how many AMI (recent of previous) was present in your population. Probably another interesting evaluation could be made on your strategy for carotid stenosis treatment: how many patients received a surgical or endovascular treatment of the carotid arteries in your population and how long before?

Response: We thank the reviewer for these comments. “Extensive rehabilitation” was performed especially for patients whose activities of daily living (ADL) were limited due to treatment for heart failure after hospitalization. Rehabilitation aims to be as close to the original ADL as possible. As a first goal, we place importance on standing (out-of-bed exercise). The result of this as an endpoint has been listed in table 2 (new table). Extensive rehabilitation is not applied to patients with high a risk for preoperative load, such as cases with LMT lesions, symptomatic cases, and unstable circulation. Additionally, patients who developed symptoms during extensive rehabilitation underwent surgery immediately. In our study, only one patient required surgery after halting extensive rehabilitation. Table 2 describes the patient’s rehabilitation period, whether they were able to stand, and whether they were able to move to the ward. Additionally, we have added a paragraph on the completion rate and a justification for rehabilitation (rehabilitation indication) to the manuscript as follows: “All non-urgent patients underwent extensive cardiac rehabilitation prior to surgery. Even patients that were admitted into the emergency room underwent cardiac rehabilitation if surgical intervention was not urgent. Additionally, we treated dental conditions, glycemic imbalances, and co-morbid treatable diseases such as carotid artery stenosis prior to cardiac surgery. Extensive rehabilitation had been especially applied for patients whose activities of daily living (ADL) were limited due to treatment for heart failure after hospitalization. Rehabilitation aims for patients to be as close to the original ADL as possible. As a first goal, we place importance on standing (out-of-bed exercise)〔5〕. Extensive rehabilitation is not applied to patients with a high risk for preoperative load, such as cases with LMT lesions, symptom cases, and unstable circulation. Additionally, patients who developed symptoms during extensive rehabilitation underwent surgery immediately.” (Page 7; line 85) and “Extensive rehabilitation had been performed in 19 (4.8%) patients. There was a significant difference between the prophylactic IABP group that underwent extensive rehabilitation (8 [9.4%] vs 11 [3.6%], respectively; p=0.041) (table 2). Seventy-nine percent (15 patients) of the patients started with in-bed exercise and eventually advanced to out-of-bed exercise (63% [5 of 8] vs 91% [10 of 11], respectively; p=0.262). Reasons for extensive rehabilitation (the reason why ADL decreased) were due to many factors including the onset of heart failure (8 patients), the onset of AMI (6 patients), temporarily use of IABP after hospitalization (3 patients), onset of AMI and heart failure (1 patient), and brain infarction (1 patient). In this study, only one patient underwent surgery with the need to halt extensive rehabilitation (table 2).” (Page 12; line 177).

5. Hodgson CL, Stiller K, Needham DM, Tipping CJ, Harrold M, Baldwin CE et al. Expert consensus and recommendations on safety criteria for active mobilization of mechanically ventilated critically ill adults. Crit Care 2014;18:658. doi: 10.1186/s13054-014-0658-y.

 We changed the sentence “all non-urgent patients underwent extensive cardiac rehabilitation prior to surgery. Even patients that were admitted into the emergency room underwent cardiac rehabilitation if surgical intervention was not urgent. Additionally, we treated dental conditions, glycemic imbalances, and co-morbid treatable diseases such as carotid artery stenosis prior to cardiac surgery.” to “preoperative extensive rehabilitation might have been something related to prevent occurring perioperative complication,” (Page 17; line 264).

 For recent AMI, the progress of all members was confirmed, and the results were listed in table 1. A recent AMI was defined as being diagnosed with AMI when hospitalized. As a treatment for stenotic lesion of the carotid artery, none of the patients in this patient group underwent a surgical operation (including stent insertion). Nine patients had stenosis of the carotid artery before surgery and were consulted for neurosurgery department at this hospital. Single photon emission computed tomography was performed. It was decided that there was little need for surgical treatment. Our policy is that if there is a time allowance for those who have surgical indications for carotid artery lesions, they will be treated first, but in this study there were no such patients. 

We added the following paragraph: “For recent AMI, the progress of all members was confirmed and the results were listed in table 1. A recent AMI was defined as being diagnosed with AMI when hospitalized. There were no differences between the two groups (12 [14.1%] vs 31 [10.1%], respectively; p=0.326). Our policy is that if there is a time allowance for those who have surgical indications for carotid artery lesions, they will be treated first, but in this study there were no such patients. As a treatment for stenotic lesion of the carotid artery and its treatment, none of the patients in this patient group underwent a surgical operation (including stent insertion). Nine patients had pointed stenosis of the carotid artery before surgery and all were consulted for neurosurgery at this hospital. Additionally, single photon emission computed tomography was performed. It was decided that there was little need for preoperative surgical treatment.”

 (Page 11; line 158).

5 Comment: What about postoperative AKI?

Response: Thank you for your comments. We added post-operative occurrence of AKI, whether dialysis was required, serum Creatinine just before surgery, and max Creatinine after surgery in tables 1 and 4. Only one patient (without prophylactic IABP) with end-stage renal disease requiring maintenance hemodialysis therapy was initiated chronic dialysis following cardiac surgery. 

We added the following sentence in the Patients and Methods section: “Acute kidney injury (AKI ) was defined by increased serum creatinine (SCr) to 0.3 mg/dl/48h or more, increased SCr by 1.5 times or more, or decreased urine output to 0.5ml/kg or less during 6 hours〔3, 4〕.” (Page 6; line 68)

“There were no differences in the post-operative occurrence of AKI between the two groups (3 [3.6%] vs 6 [2.1%], respectively; p=0.326). However, patients without prophylactic IABP required more dialysis following cardiac surgery (0 of 3 [0%] vs 6 of 7 [86%], respectively; p=0.033). Only one patient (without prophylactic IABP) with end-stage renal disease requiring maintenance hemodialysis therapy initiated chronic dialysis following cardiac surgery.” (Results section:Page 13; line 187)

3. Kellum JA, Bellomo R, Ronco C, Mehta R, Clark W, Levin NW. The 3rd International Consensus Conference of the Acute Dialysis Quality Initiative (ADQI). Int J Artif Organs. 2005;28:441-4.

4. Creatinine Kinetics and the Definition of Acute Kidney Injury. Sushrut SW, Joseph VB. J Am Soc Nephrol: 2009;20: 672–679.

6 Comment: There is no mention about conduction and completeness of FU.

Response: We thank the reviewer for their comments. The follow-up results after discharge were confirmed on the medical records of this hospital, related hospitals, or via telephone. In accordance with the reviewer’s comment, we have added a new sentence as follows: “There was a 93% postoperative follow-up rate over an average of 42.5 months (± 42.6). The follow-up results after discharge were confirmed on the medical records of our hospital, related hospitals, or via telephone.” (Page 12; line 183)

7 Comment: In my opinion an actuarial survival estimated by KM analysis has very little sense in a period of observation of 30 days. It should be made on the first year at least.

Response: Thank you for your suggestion. In agreement with the reviewer, we have incorporated the suggestion into our paper. We examined the one-year survival rate between the two groups and included one-year outcomes in Figure 2. We added a new sentence in the Results section as follows, “The 12-months survival curves in the 85 high-risk patients who received prophylactic IABP compared to the 308 non-high risk patients who did not receive prophylactic IABP were similar (p=0.846) (Fig 2).” (Page 14; line 208)

8 Comment: The topic is hot and the consensus on the benefit of prophylactic IABP is not so widespread. So in my opinion could be important to emphasize the open question in the discussion section and the references must be enlarged. Furthermore the recent article from Rampersad et al is not a RCT but a metaanalysis.

Response: We appreciate these insightful comments. We apologize for the incorrect description. We have corrected the statement in the Discussion section as follows: “A recent meta-analysis (MTA) reported that prophylactic IABP use reduced mortality and MACCE in high-risk CABG patients〔10〕” (Page 7; line 9)

 In agreement with reviewer #1, the consensus of prophylactic IABP is not so widespread and may need further consideration in the future. In this study, we could mention that the use of prophylactic IABP did not cause major complications and the patients’ mortality rates were not high. As proposed, the discussion was edited to address the inconsistencies on the beneficial effects of IABP. Additionally, a reference was added. 

We added a new sentence in the Discussion section as follows: “Currently, the consensus on the benefit of prophylactic IABP is not so widespread.” (Page 14; line 215), “Multiple studies reported the benefit of prophylactic IABP [1,2, 6-8], but it had not been established as the gold standard for high-risk CABG. One of the reasons was complications.” (Page 14; line 219), “Although we were unable to determine the advantages of using prophylactic IABP, some papers report good results.” (Page 15; line 237) and “Jiayang et al showed that preoperative prophylactic IABP reduced the incidence of CABG-associated AKI〔15〕. Our results also showed that postoperative dialysis was significantly less in the prophylactic IABP group (0 of 3 [0%] vs 6 of 7 [86%], respectively; p=0.033). The utility of prophylactic IABP is still open for further study, but early studies indicate that postoperative results may be improved. However, further research is warranted to demonstrate the beneficial effects of prophylactic IABP use.” (Page 16; line 248)

15. Wang J, Yu W, Gao M, Gu C, and Yu Y. Preoperative Prophylactic Intraaortic Balloon Pump Reduces the Incidence of Postoperative Acute Kidney Injury and Short-Term Death of High-Risk Patients Undergoing Coronary Artery Bypass Grafting: A Meta-Analysis of 17 Studies. Ann Thorac Surg 2016;101:2007–19

9 Comment: The tables are very difficult to read, please reorganize and simplify 

Response: Thank you for your suggestion. In accordance with the reviewer’s comment, the tables have been reorganized.

Reviewer #2

1 Comment: It is a very interesting field because IABP seems has to be abandoned by cardiological guidelines but still it has been use consistently in cardiac surgery. As you underlined the cohort of patients in not very wide and is not possible to be definitive on the prophylactic IABP with this numbers

Where do you allocate the patient in the preoperative period? Has to be at least an HDU. Has this time counted in the ICU admission stay? In my institution we insert the IABP after the induction in OR.

Response: Thank you for your insightful comments. All patients who received a prophylactic IABP entered the ICU and were waiting. “ICU stay” was not counted as ICU admission stay during the preoperative period.

We have added a new sentence as follows: “All patients who received a prophylactic IABP entered the ICU and were waiting.” (Page 6; line 75) “ICU stay” was not counted as ICU admission stay during the preoperative period.” (Page 13; line 194)

2 Comment: It seems that the method you have used to detect high risk patients is not working properly since there is a consistent use in not high risk group.

Plus is very interesting that the high risk LVEF, despite is significantly different from the other group, is 49%.

The last consideration is on the weaning time. Have you got any standard parameter to consider to asses the possibility to wean? Do you you use levosimendan?

Response: Thank you for your insightful comments. The patient selection was pointed out by reviewer #1. In order to make the selection as appropriate as possible, we have re-examined only with isolated CABG, except for combined CABG. All of these values were listed in tables 1, 3, and 4. Three hundred ninety-three patients were included in the isolated CABG group. All statistical analyses were repeated. All applicable locations and numbers have been changed. We added the sentence in the Patients and Methods section to the following. “at Nihonkai General Hospital, and involved 471 unique patients from December 2005 to December 2017 who underwent isolated or combined CABG. Of those, 393 patients underwent isolated CABG and were included for the analysis.” (Page 5; line 51)

　IABP weaning was dependent on stabilization of circulatory dynamics. Weaning was performed when diuresis was present and there was warm in peripheral sensation. However, there was no numerical target. Also our facility did not use levosimendan as an aid to weaning. 

We have added a new sentence as follows: “IABP weaning was dependent on stabilization of circulatory dynamics. Weaning was done when diuresis was present and there was warm in peripheral sensation. However, there was no numerical target.” (Page 7; line 82)

---

## [Editor Report · Decision Letter 1]

10 Oct 2019

The use of prophylactic intra-aortic balloon pump in high-risk patients undergoing Coronary Artery Bypass Grafting

PONE-D-19-19063R1

Dear Dr. Nakmaura,

We are pleased to inform you that your manuscript has been judged scientifically suitable for publication and will be formally accepted for publication once it complies with all outstanding technical requirements.

With kind regards,

Andrea Ballotta

Academic Editor

PLOS ONE

Additional Editor Comments (optional):

Thank you again for your efforts, i deem your paper suitable for publication.
---

## [Editor Report · Acceptance letter]

21 Oct 2019

PONE-D-19-19063R1 

The use of prophylactic intra-aortic balloon pump in high-risk patients undergoing Coronary Artery Bypass Grafting 

Dear Dr. Nakmaura:

I am pleased to inform you that your manuscript has been deemed suitable for publication in PLOS ONE. Congratulations! Your manuscript is now with our production department. 

With kind regards,

on behalf of

Dr. Andrea Ballotta 

Academic Editor

PLOS ONE